



# A Random Forest approach to quality-checking automatic snow-depth sensor measurements

Giulia Blandini[1,2], Francesco Avanzi[1], Simone Gabellani[1], Denise Ponziani[1], Hervé Stevenin[3], Sara Ratto[3], Luca Ferraris[1,2], and Alberto Viglione[4]

[1]CIMA Research Foundation, Savona, Italy
[2]DIBRIS, University of Genoa, Genova, Italy
[3]Centro Funzionale Valle D'Aosta, Aosta, Italy
[4]Department of Environment, Land and Infrastructure Engineering, Politecnico di Torino, Turin, Italy

**Correspondence:** Giulia Blandini (giulia.blandini@edu.unige.it)

**Abstract.** State-of-the-art snow sensing technologies currently provide an unprecedented amount of data from both remote sensing satellites and ground sensors, but their assimilation into dynamic models is bounded to data quality, which is often low – especially in mountain, high-elevation, and unattended regions where snow is the predominant land-cover feature. To maximize the value of snow-depth measurements, we developed a Random Forest classifier to automatize the quality assurance/quality control (QA/QC) procedure of near-surface snow depth measurements collected through ultrasonic sensors, with particular reference to differentiate snow cover from grass or bare ground data and to detecting random errors (e.g., spikes). The model was trained and validated using a split-sample approach of an already manually classified dataset of 18 years of data from 43 sensors in Aosta Valley (north-western Italian Alps), and then further validated using 3 years of data from 27 stations across the rest of Italy (with no further training or tuning). The F1 score was used as scoring metric, being it the most suited to describe the performances of a model in case of a multiclass imbalanced classification problem. The model proved to be both robust and reliable in the classification of snow cover vs. grass/bare ground in Aosta Valley (F1 values above 90%), yet less reliable in rare random-error detection, mostly due to the dataset imbalance (samples distribution: 46.46% snow, 49.21% grass/bare ground, 4.34% error). No clear correlation with snow-season climatology was found in the training dataset, which further suggests robustness of our approach. The application across the rest of Italy yielded F1 scores on the order of 90% for snow and grass/bare ground, thus confirming results from the testing region and corroborating model robustness and reliability, with again a less skillful classification of random errors (values below 5%). This machine learning algorithm of data quality assessment will provide more reliable snow ground data, enhancing their use in snow models.



## 1    Introduction

Snow plays a key role in shaping the dynamics of the hydrological cycle, influencing streamflow as well as surface and ground water storage availability in terms of quantity, quality, and timing (Dettinger, 2014). Snow depth measurements and related Snow Water Equivalent data (SWE) provide insightful knowledge, exploitable for water management, hydrological forecasting and emergency preparedness (Hartman et al., 1995). Recent analyses prove that a significant reduction of streamflow is often a direct consequence of a reduction of precipitation as snow, exacerbated by temperature increase (Berghuijs et al., 2014). In this

framework, snow droughts severely affect the hydrology cycle (Harpold et al., 2017) leading to hydrological droughts (Toreti et al., 2022). Additionally, the snow cover and ice cover are key climate change indicators, especially because the high albedo and low thermal conductivity of snow cover strongly affect the global radiant energy balance and the atmospheric circulation (Flanner et al., 2011). A decline in snowfall and snow cover on the ground not only affects water supplies, but it also alters the equilibrium of wildlife and vegetation as well as transportation, cultural practices, travel, and recreation for millions of people

(Bair et al., 2018).

Assessing the implication of snow-driven hydrological processes on the timing and quantity of streamflow and precipitation events helps in forecasting for water management, dealing with water security and water related vulnerability, and enforcing the development of a sustainable water resource carrying capacity, especially considering the shift in water balance caused by climate change (Maurer et al., 2021). In this framework, physics models are often used to support engineers, scientists, and

decision makers in real world hydrological operations.

Contemporary environmental technologies have made it possible to easily gain new information, even in real time, with an increasing quantity of data made available from remote sensing satellites, and more sophisticated ground sensors. However, high resolution data of snow come with a variety of noise sources that make a Quality Assurance and Quality Control (QA/QC) indispensable to use such data in snowpack modeling (Avanzi et al., 2014; Bavay and Egger, 2014). A recurring case in this

context are snow-depth data, with two frequent noise categories: (1) snow vs. grass ambiguity, due to snow depth ultrasonic sensors detecting not only snow cover but also plant/grass growth in spring and summer (Vitasse et al., 2017), and (2) random errors (e.g., spikes, anomalous data points that protrude above or below an interpolated surface).

Traditionally, in the field of snow cover and snow depth monitoring, quality assurance/quality checking procedures have been carried out by visual inspection, heavily depending on subjective expert knowledge (Robinson, 1989). While expert-knowledge

QA/QC is arguably the most reliable approach to data processing, these practices are not easily reproducible or transferable, and highly time-consuming (Fiebrich et al., 2010). In this context, QA/QC with regard to grass detection is often based on static climatological or minimum-snow-depth thresholds, while random errors are generally detected based on maximum-snow-depth thresholds or criteria based on signal variance (Avanzi et al., 2014). An exception in this regard is the approach implemented by the Swiss MeteoIO for grass detection, which however requires information on surface snow temperature, ground surface

temperature and radiation (Bavay and Egger, 2014).

In view of this knowledge gap, Jones et al. (2018) highlight the burden of subjectivity that may affect overall data quality and comparability, stressing how even expert scientists are not immune to mistakes, especially if performing recurrent unguided





quality checking procedures. As explained by Schmidt et al. (2018), automatic environment data quality control literature is still fragmented, with heterogeneous applications. It is clear then the necessity for a quality checking procedure, that ought to
be defined through common and iterable guidelines to guarantee repeatability and consistency (Jones et al., 2018).

Considering the ever growing volume of data and the limitations arising from traditional QA/QC procedures, here we follow intuitions from Schmidt et al. (2018) and propose the use of machine learning to automatically quality check high-resolution snow-depth sensor data from ultrasonic sensors. The choice of machine learning was driven by its efficiency in dealing with big data sets and as a valid reinforcement of traditional analytic tools (Ferreira et al., 2019). Moreover, machine learning
techniques may also be able to handle different data formats more easily that traditional statistical tools, while they better deal with combination of features that are a-priori unknown to the developer (Zhong et al., 2021).

We trained and validated our algorithm using as training dataset an already classified pool of 18 years of hourly data from 43 snow-depth sensors in Aosta Valley. We then expanded the validation by applying the final algorithm over 3 years of independent data from 27 stations across the rest of Italy (no further tuning in this case), as a pilot case study to assess the
applicability of this algorithm to larger and more heterogeneous domains. This research thus answered three questions: (i) what is the accuracy of a Random Forest classifier algorithm in automatically performing QA/QC of near-surface snow depth observations? (ii) is the approach transferable to untested regions and, if so, what is the potential drop in performance? (iii) how do meteorological conditions influence model performance and the Random-Forest decision process?

This paper is organized as follows. Section 2 describes the dataset used to train and test the Random Forest algorithm.
Section 3 describes the methodology followed to develop such algorithm. Finally, Section 4 provides an analysis of the results, while Section 5 discusses the main findings and implications of our work.

## 2  Data

To develop, test and validate the algorithm, two different datasets were used: a dataset with 18 years of already classified snow-depth data at 43 locations from Aosta Valley, which was used as intensive study domain to develop the algorithm, and 3
years of data from 27 snow-depth sensors across the rest of Italy, which were used to test the generalization and transferability of the algorithm in time and space.

### 2.1  Aosta Valley data

Aosta Valley is located in the northwest Italian Alps (Figure 1). The region includes some of the highest peaks in the Alps (Mont Blanc - 4808 m asl, Monte Rosa - 4634 m asl, Mount Cervino -4478 m asl, and Gran Paradiso - 4061 m asl). While some of
these peaks, such as Mont Blanc, are inner-Alpine, others, such as Monte Rosa and Gran Paradiso, overlook the Pianura Padana and thus are more exposed to maritime conditions (Sturm and Liston, 2021). This generates marked precipitation-regime discrepancies, hence climatic differences and different snow regimes, across the region (Avanzi et al., 2021b). Despite being a comparatively small mountainous region, the climate variability and the abundance of already classified snow measurements were the reasons that led us to the choice of this area as training domain.





The Aosta-valley data-set consists of hourly snow depth measurements from 43 ultrasonic sensors (precision in the order of a
few cm, Figure 1, (Ryan et al., 2008)), provided by the regional Functional Center of the National Civil Protection Department.
The period of record goes from August 2003 to September 2021, thus covering a variety of snow seasons across 18 years of
data (Avanzi et al., 2022b). The elevation range of these sensors goes from 545 m to 2842 m asl, with an average elevation of
2007 m asl that is representative of average elevations across the Italian Alps where the bulk of sensors are located (Avanzi
90     et al., 2021b).

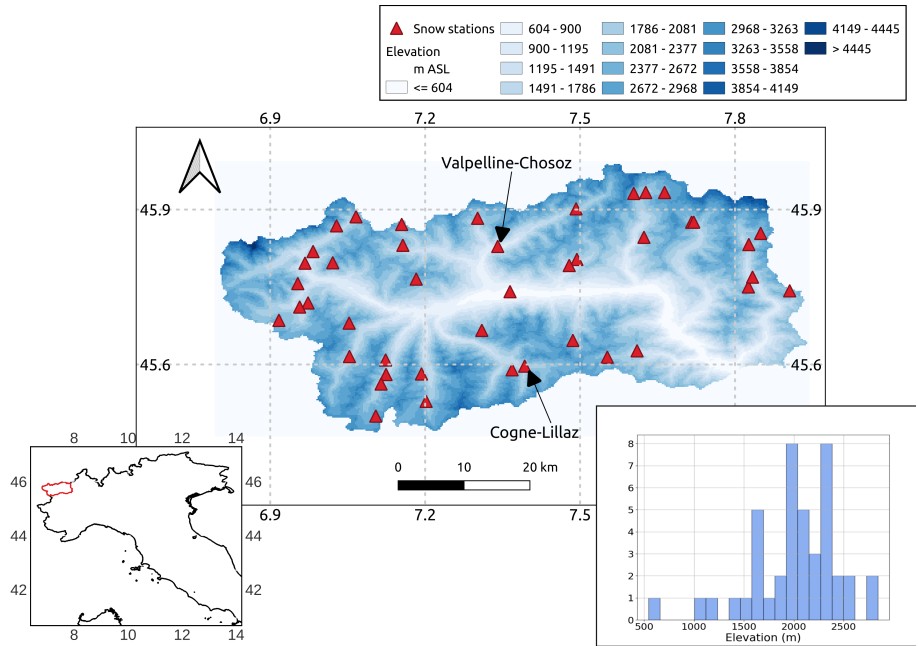

**Figure 1.** Considered snow-depth sensor data across Aosta Valley (see the bottom-left corner for the location of this study region in Italy).
The two snow-depth sensors of Valpelline-Chosoz and Cogne-Lillaz were used in section 4.3. The histogram in the bottom-right corner of
this figure reports the frequency distribution of the elevation of the Aosta-Valley sensors.

     Each data record in this dataset was subject to visual screening by expert hydrologic forecasters during periodical QA/QC
manual data processing, with the goal of discriminating random and systematic errors from actual snow depth measurements.
This manual processing follows well established practice in the field, including crosschecking with concurrent weather (e.g.,
air temperature, precipitation, relative humidity) and nearby sensors (Avanzi et al., 2014, 2020a). As a result, each datapoint
95     came with a quality code (Table 1): data with code 0 or 1 are valid snow cover data; codes 2 or 4 are for missing data
reconstructed from trends or aggregated from different time resolutions; codes 8 and 16 are grass or bare ground; code 32
denotes reconstructed grass data; codes 64 to 256 denote a variety of flags for random and instrumental errors; codes 1024-
1032 refer to data classified as invalid after a preliminary procedure based on fixed thresholds (introduced in 2018). While
the dataset includes some reconstructed data, these are only 0.03% of the whole dataset, which means they do not affect our
analyses.





In this work, we reduced the number of classes to 3 by aggregation: correct snow depth, identified with code "0", grass or bare ground, identified with code "1", and random errors, identified with code "2".

| Code | Data type | % on total | Code | Data type |
|---|---|---|---|---|
| 0 or 1 | Valid snow data | 46.43% | | |
| 2 | Qualitatively (aggregated) valid snow data | <0.01% | 0 | Snow data |
| 4 | Reconstructed missing snow data | 0.03% | | |
| 8 or 16 | Grass/bare ground data | 49.20% | 1 | Grass/bare ground data |
| 32 | Reconstructed missing Grass/bare ground data | 0.01% | | |
| 64-72 | Random error ,invalid data | 3.95% | | |
| 128 | Calibration error | 0.02% | 2 | Errors |
| 256 | Maintenance error | 0.03% | | |
| 1024-1032 | Rejected data based on climatological thresholds | 0.34% | | |

**Table 1.** Snow depth data classification system developed by the Functional Center of Aosta Valley.

## 2.2 Other Italian Data

The validation dataset across the rest of Italy comprises hourly data from 27 ultrasonic depth sensors, randomly chosen among the ~300 Italian automatic snow-depth sensors available outside Aosta Valley. These 27 snow-depth sensors were chosen based on a geographical-diversity criterion to guarantee heterogeneity, especially with regard to the Aosta Valley data (Figure 2). This second dataset include data from 3 years: 2018, 2020 and 2022, which were chosen due to their significantly different accumulation patterns (deep snowpacks in 2018, somewhat average snowpacks in 2020, and extraordinarily low snowpacks in 2022, see Avanzi et al. (2022a)). No prior processing was available for these data, thus we proceeded with our own manual classification to assign codes as in Table 1. The procedure included visual screening, checks on seasonality to detect snow vs. grass, and a comparison with measurements from nearby sensors (Avanzi et al., 2014, 2020a).

## 3 Methods

### 3.1 Random Forest: background

Among all machine learning approaches, we chose Random Forest due to its benchmarking nature as well as its simplicity of use (Tyralis et al., 2019), as proven by an increasing number of studies proving the effectiveness of Random Forest as classifier or regressor algorithm. For instance, Desai and Ouarda (2021) developed a flood frequency analysis based on Random Forest, which proved to be equally reliable but more efficient than more complex models; Park et al. (2020) developed a Random Forest classifier for sea ice, using Sentinel-1 data; Random Forest proved to be efficient in big data environments (Liu, 2014); recently, Ponziani et al. (2023) proved the efficiency of Random Forest over other machine learning algorithm,developing a



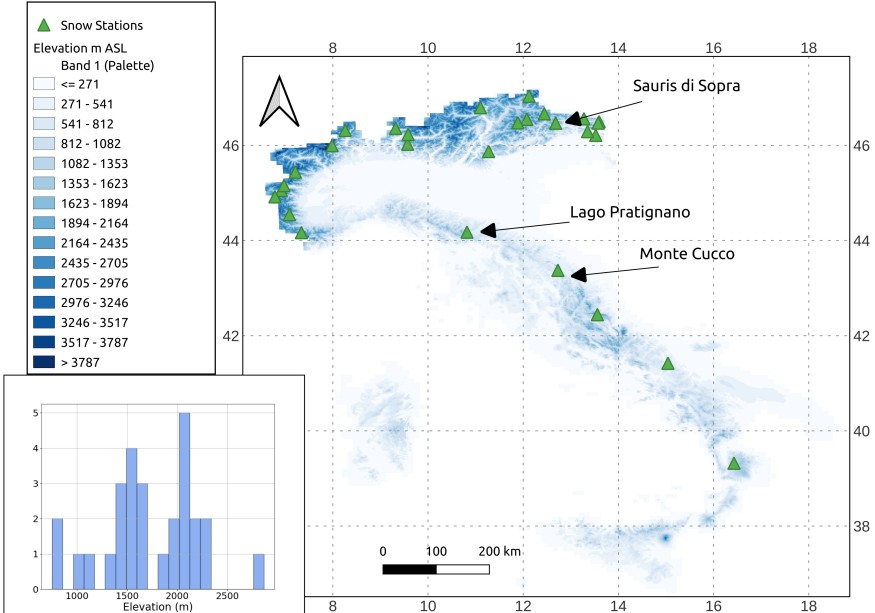

**Figure 2.** Considered snow-depth sensor data across the rest of Italy. Bottom left: frequency distribution of the elevation of these sensors. Three black arrow indicates the location of three snow-depth sensor used in section 4.5 .

predictive model for debris flows that could be experimentally implemented in the existing early warning system of the Aosta valley. In particular, the algorithm object of the present study is a Random Forest classifier, an ensemble classifier based on bootstrap aggregation and random features selection.

A Random Forest is an ensemble of decorrelated decision trees that are let growing and voting for the most popular class (Breiman, 2001). The growth of each tree in the ensemble is governed by randomness, proved to be a performance enhancer.

Randomness is given by two randomization principles: bagging and random feature selection. According to the bagging principle, a large number of relatively uncorrelated trees, each built using a split sample of $n$ dimensions retrieved from the entire training dataset of size $m$, operate as a committee; this ensemble is proven to outperform any of the individual constituent trees. Therefore, the class definition, made by averaging the scores of each tree, is mildly affected by the weight of misclassification done by less performant trees. Furthermore, instead of splitting a node searching the most important feature (i.e., predictor),

a Random Forest uses the best one among a random subset of features, performing random feature selection, thus increasing the performances. Randomness injection minimizes correlation across trees and reduces variance and overfitting, increasing stability (Breiman, 2001). Our algorithm was implemented using Scikit-learn Version 0.20.1, a Python software programming platform, using the class RandomForestClassifier.





### 3.2 Random Forest Development

To train the Random Forest, we used the Aosta Valley classified dataset. Based on data frequency (Figure 3), this is a typical imbalanced dataset where class distribution is skewed or biased towards one or few classes in the training dataset (Kuhn et al., 2013). In this framework, data belong either to majority or minority classes. The majority classes are the classes with a larger number of observations, while the minority classes are those with comparatively few observations. In this case, the number of data classified as random errors (code 2) is significantly lower than the number of data from category 0 (snow height) or
category 1 (grass/bare ground). Thus, classes 0 and 1 were defined as majority class, while class 2 was defined as minority class.

**Population subdivision into classes**



**Figure 3.** Aosta-Valley data subdivision into classes.

Class imbalance can severely affect the classification performance (Ganganwar, 2012; Ramyachitra and Manikandan, 2014), and therefore requires a pre-processing strategy. To this end,acknowledging the work of (Ponziani et al., 2023) in which no clear evidence of out-performance of any strategy, we performed an oversampling of the minority class, by selecting examples
to be duplicated and then added to the training dataset and using the class RandomOversampler from the package Imbalanced-learn version 0.8.1. To decrease the computational effort that may have stem from this oversampling procedure (Branco et al., 2016), a representative sample of $1.3 \times 10^6$ measurements was taken from the entire dataset prior oversampling of the minority class for Random Forest training. This sample was proven to be representative of the entire dataset distribution (approximately $5.5 \times 10^6$ datapoints), by performing a two-sample Kolmogorov-Smirnov test, with a significance level equal to 0.05. After the
oversampling procedure, a sample of $1.9 \times 10^6$ over-sampled measurements was used to train the Random Forest. From the remaining, not oversampled dataset, an independent test sample of $4.8 \times 10^5$ measurements was randomly selected. As a result, a train-test split share of 80% training 20 % test was used, in agreement with current standards in machine learning problems (Harvey and Sotardi, 2018).



When dealing with an imbalance classification, standard evaluation criteria focusing on the most frequent classes may lead to misleading conclusions, because they are insensitive to skewed domains (Branco et al., 2016). For example, accuracy,which is defined as the number of correct predictions over total number of predictions,that is frequently used metrics for classification problems, underestimates the importance of the least represented classes when compared with the majority classes, as it does not take into account data distribution. Adequate metrics need to be used not only for model validation, but also for model selection, given that accuracy scores may ignore the difference between types of misclassification errors, as they seek to minimize the overall error. A good metric for imbalance classification must consider overall data distribution, giving at least same importance to misclassification in both majority and minority classes.

In this paper, we thus used the F-measure (Van Rijsbergen, 1979), that is, the harmonic mean of precision(measure of exactness), defined as the number of true positives divided by the total number of positive predictions, and recall (measure of completeness), defined as the percentage of data samples that a machine learning model correctly identifies as belonging to a class of interest out of the total samples for that class.

The harmonic mean is the reciprocal of the arithmetic mean and tends to mitigate the impact of large outliers, while aggravating the impact of small ones since it tends strongly toward the least represented elements.

F$\beta$ (the so-called F-measure) is defined as:

$$F_\beta = \frac{(1+\beta)^2 recall * precision}{\beta^2 recall + precision} \tag{1}$$

We set $\beta = 1$ to give equal importance to precision and recall. Macro-averaged values of F1 score, regardless of the weight of each class, were considered to analyse the model performances. The performances of the trained Random Forest algorithm were tested on the 20% test dataset, using the model in prediction and comparing model's classification with that by the expert forecasters. Validation was also performed by applying the final algorithm on the 3 years of data from the rest of Italy (Section 2.2).

We chose as candidate predictors (features) of our Random Forest a collection of meteorological, topographic, and temporal variables that are known to influence snow accumulation and melt, thus mimicking the decision process made by experts when assigning a classification code. These features include snow-depth values themselves, elevation, aspect, concurrent air temperature, radiation, total precipitation, wind speed, relative humidity, and the day of the year. Feature values were extracted for each datapoint in both the Aosta Valley and the rest-of-Italy samples, using available geographic information and weather maps operationally developed by CIMA Research Foundation (see Avanzi et al. (2021a) for Aosta Valley and Avanzi et al. (2022a) for other Italian data).

A feature importance analysis was also performed. Importance was calculated using the attribute "feature importance" of the class RandomForestclassifier in sklearn.ensemble (Pedregosa et al., 2011). The ranking is driven by each feature contribution to a decrease in impurity over trees.

A set of hyper-parameters were optimized through a combination of automatic, random searching and further manual tuning to reduce overfitting, yet ensuring good F1 score and reliable training time on Aosta Valley dataset. The parameters that were





tuned in this work were the number of estimator (namely, the number of trees in the forest), the maximum depth (namely, the maximum number of levels in each decision tree), the minimum sample leaf (namely, minimum number of data points placed
in a node before the node is split), and the minimum sample split (namely, minimum number of data point needed to split an internal node). Others default hyper-parameters were not modified.

Finally, we mapped classification results as a function of feature values to shed light on the decision process taken by the Random Forest in classifying snow vs. grass/ground vs. random errors and how they relate with the original classification by operational forecasters.

## 195  4   Results

### 4.1   Training and test performances: Aosta Valley

The macro-averaged F1 score of classification for the Aosta Valley dataset during testing phase was 0.96, with a precision value of 0.97 and a recall of 0.95 (Fig. 4 panel a). In details, the Random Forest scored 0.99 in both precision and recall for the classification of snow data. In the classification of grass/bare ground, recall was maximum (1), with precision of 0.99. Lower
values were obtained in the classification of random errors, with recall of 0.86 and precision of 0.93, resulting in a F1 score of 0.89. Most of snow depth data and grass/bare ground data were correctly classified(45.94% /46.45 % and 49% / 49.19%), while a comparatively large sample of error data that was miss-classified as snow (0.50% / 4.43% (Fig. 4 panel b). Overall, the model resulted in being equally precise and robust in snow and grass/bare ground classification, while the precision and recall of the random-errors class were lower compared to the other two classes (F1 score for snow and grass/bare-ground classes of
0.99 and F1 score for random error classes of 0.89). As a whole, the model tested in Aosta Valley proved to be slightly more precise than robust (precision 0.97 , recall 0.95).

In order to identify recurring patterns in snow cover and grass cover classification during the hydrological year, we visually screened results of the Random Forest classifier for all data and hydrological years Fig.5 reports examples for two snow-depth sensor locations (October 2016 to September 2017), which were randomly selected from the entire pool of 43 snow-depth
sensors along 18 years of Aosta Valley domain. Note that we removed samples used for the Random Forest training. We found an expected tendency of the Random Forest to missclassify snow as grass/bare ground during transitional periods at the beginning and at the end of the snow season (Fig.5 pan A2), especially when snow-cover and grass height are comparable (Fig.5 pan B2 and B3). Moreover, The Random Forest sometimes misinterprets settling during snow period.

### 4.2   Model configuration

The best set of parameters for the development of the Random Forest resulted in a number of estimator equals to 500, a maximum depth of 40, a minimum sample leaf equals to 1, a minimum sample split equals to 2. The choice of the best set of features was initially driven by the F1 macro averaged obtained on the test set (Table 2, features combination sets from T1 to T7); then, training time was also considered as a discriminant (+10 minutes for T6 compared to T7). Hence, the set



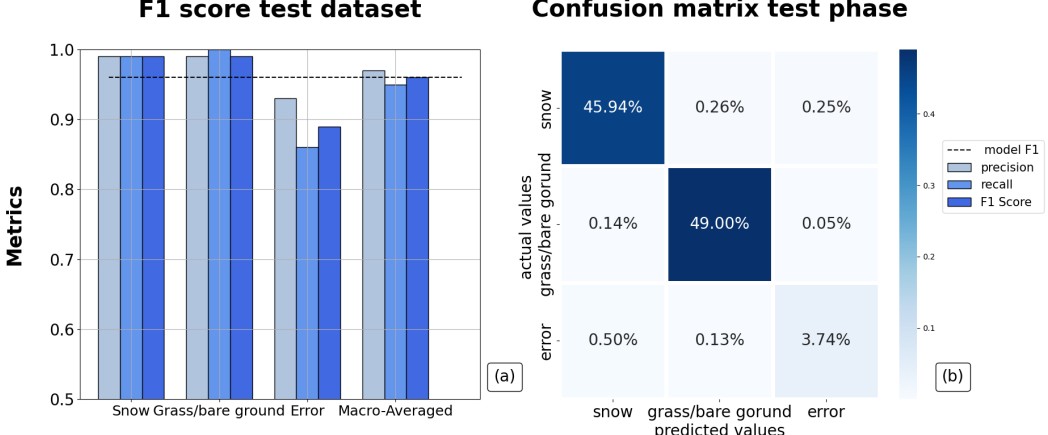

**Figure 4.** Left: model performances in prediction mode for the test dataset in Aosta Valley. Each set of columns reports the values of precision, recall, and F1 score for the three classes, while the last group on the right shows the macro-averaged values referred to the Random Forest performances as a whole. The black dashed line is a reference for the macro averaged F1 score of the Random Forest. Right: confusion matrix.

of feature selected as the best consisted in: the snow depth record measured by the snow-depth sensor, elevation, aspect, concurrent air temperature, incoming shortwave radiation, cumulative precipitation, relative humidity, and the day of the year to capture seasonality (Table 2, set T7). Regarding elevation and aspect, previous studies have shown that geographic location and elevation indeed contribute to improving machine-learning model performance (Bair et al., 2018).

| Features | T1 | T2 | T3 | T4 | T5 | T6 | T7 |
|---|---|---|---|---|---|---|---|
| Snow height | ✓ | ✓ | ✓ | ✓ | ✓ | ✓ | ✓ |
| Aspect | ✓ | ✓ | ✓ | ✓ | ✓ | ✓ | ✓ |
| Elevation | ✓ | ✓ | ✓ | ✓ | ✓ | ✓ | ✓ |
| Air temperature | ✓ | ✓ | ✓ | ✓ | ✓ | ✓ | ✓ |
| Radiation | ✓ | | ✓ | ✓ | ✓ | | ✓ |
| Relative humidity | ✓ | ✓ | | ✓ | ✓ | | ✓ |
| Cumulative precipitation | ✓ | ✓ | ✓ | | ✓ | | ✓ |
| Day of the year | | | | | | ✓ | ✓ |
| Wind velocity | ✓ | | | | | | |
| **F1 score** | 0.84 | 0.87 | 0.85 | 0.86 | 0.93 | 0.95 | 0.96 |

**Table 2.** F1 scores for a variety of tests used to identify the best feature combination for the Random Forest Algorithm. T7 was then selected as best option in terms of features.



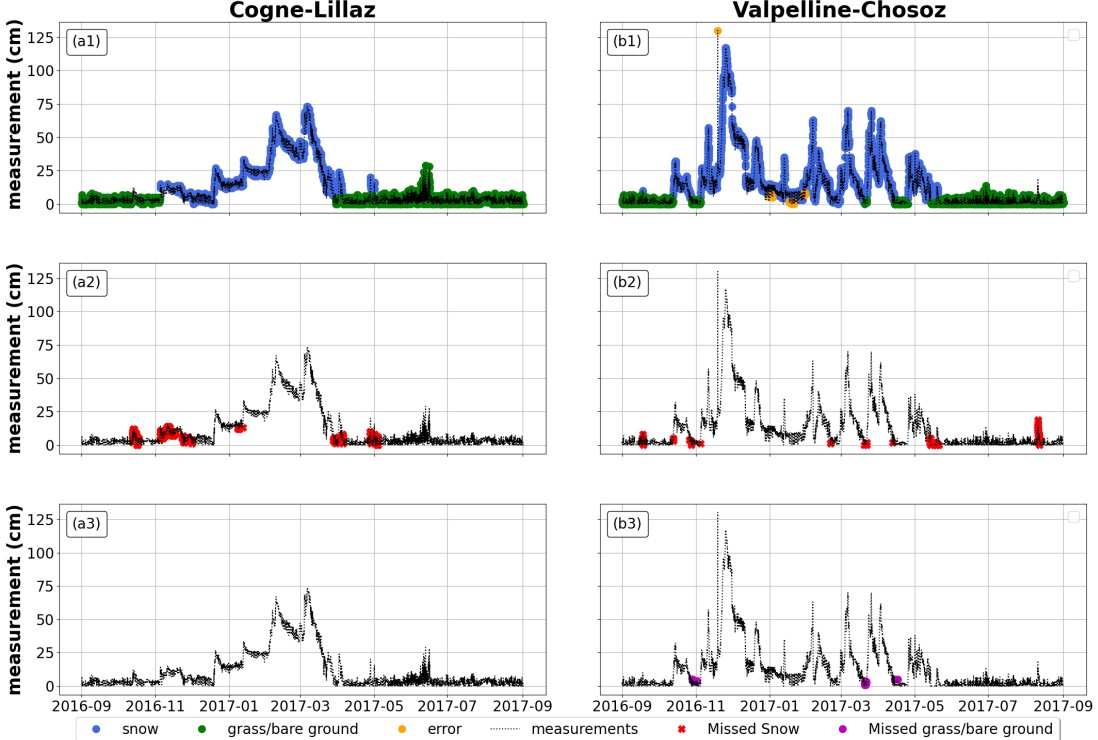

**Figure 5.** Application of Random Forest on 2 Aosta Valley snow-depth sensors locations from October 2016 to September 2017. The first row displays the samples of snow height, grass/bare ground and error correctly classified by the model.In Blue correctly classified snow sample, in green correctly classified grass sample, in orange correctly classified error. The second row shows miss-classified snow height in red and the third row reports miss-classified grass/bare ground samples in purple. Data are refereed to an hydrological year.

Feature importance (Fig.6) suggested that measured snow-depth itself (regardless of whether is represents actual snow depth, grass, bare ground, or random errors) was the most important feature in our Random Forest, followed by the day of the year, 225 air temperature, and aspect. Radiation, relative humidity, and elevation scored similarly, while total precipitation was the least important feature. Feature importance results followed a somewhat intuitive ranking, similar to human perception. For example, the model gave high importance to snow depth, likely to replicate the concept of a "plausible range" of snow depth as opposed to grass, bare ground, or random errors. Seasonality (expressed as day of the year and air temperature) was the second most influencing factor, likely to mimic the concept of a "plausible" period for snow on the ground. Aspect and elevation were less 230 influential, which is likely because of the comparatively small size of the Aosta Valley study region.



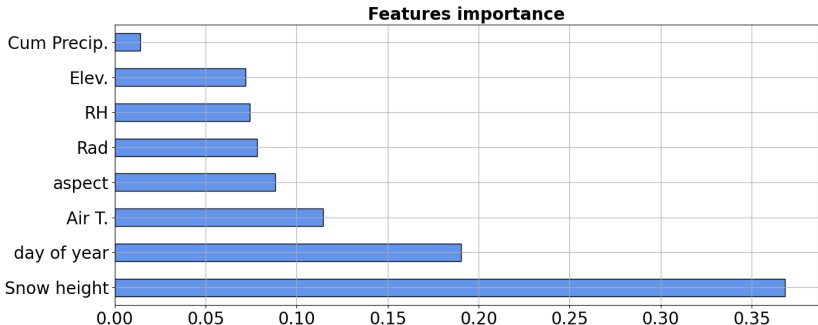

**Figure 6.** Feature importance for the Random Forest classification procedure in Aosta Valley. The dimensionless values, along the x-axis, sum up to 1; the higher the value the more important the feature is in the definition of the class. In particular: Cum, Precip. = cumulative precipitation, Elev = elevation, Rh = relative humidity, Rad = radiation, air T = air temperature

### 4.3 F1 correlation with annual climate

Annual mean feature values showed low or negligible correlation coefficient with the annual F1 score (Fig. 7, with removal of training data points). All correlation coefficients were statistically tested and no correlation was found (p-value between -0.21 and 0.40 for all the features).

### 4.4 Mapping the decision process

Analysis of the Random Forest decision process highlighted consistency with the classification procedure by expert forecasters, as well as agreement with the expected decision process behind the human made classification, despite a general underestimation of the number of random error samples (Fig.8).

The frequency of data classified as snow decreased with increasing temperature, as expected and in agreement with the original expert classification (Fig.8 pan. a1). Simultaneously, the frequency of data classified as grass/bare ground increased with temperature (Fig.8 pan. a2), again as expected due to progressive melt and snow disappearance as temperature increases. Regarding random errors, the Random Forest underestimated their frequency up to 10 ˚C, while automatic and human-made classifications were more comparable in frequency above that temperature threshold (Fig.8 pan. A3).

Considering the day of the year, most snow classifications occurred at the beginning and at the end of the calendar year (thus, in winter); this proved to be consistent between the Random Forest and the human classification (Fig.8 pan. B1), with then a shift towards the grass/bare ground class in summer (Fig.8 pan. B2). Overall, we found a underestimation of random error samples throughout the year, especially in the first 150 days of the year (Fig.8 pan. B3).

The number of data classified as snow progressively increased with elevation (Fig.8 pan. c1), while the number of classification as grass/bare ground samples decreased with elevation (Fig.8 pan. c2), consistently between the Random Forest and



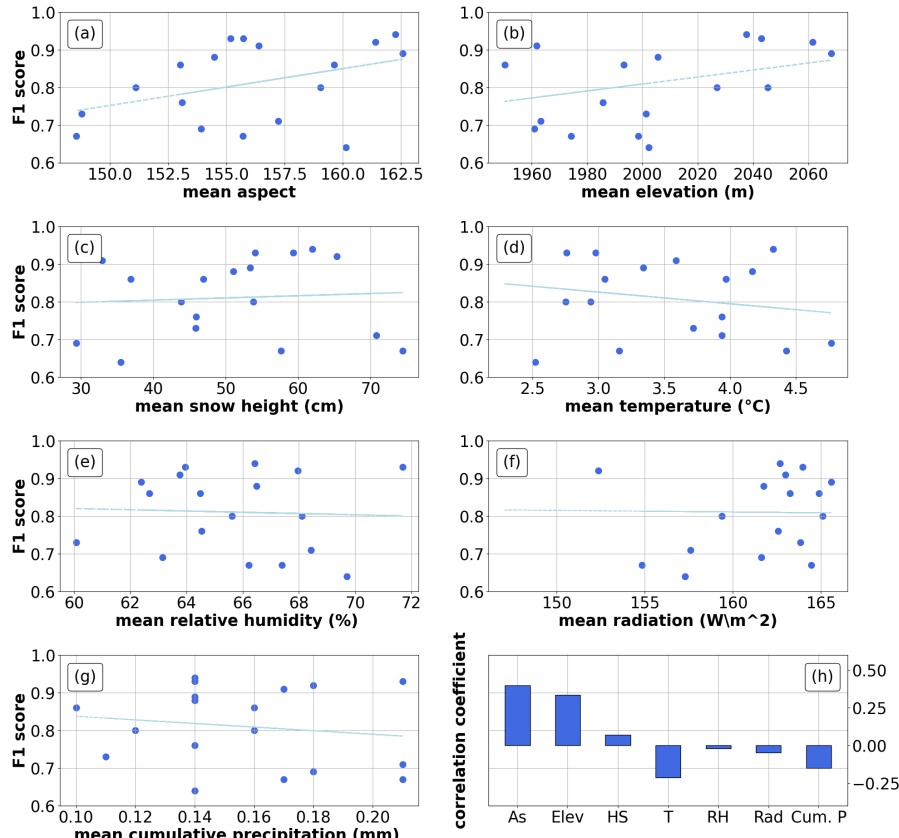

**Figure 7.** Annual F1 score correlated with mean annual feature values. The y-axis reports the F1 score macro averaged for each year, while the x-axis shows the values of annual mean for each feature.The blue straight line indicates a linear regression . The last plot indicates the correlation coefficient between single features and F1 score.

the original dataset. The frequency of random-error classifications generally matched the human classification, except for an underestimation around 2500 m (1% of miss-classified samples) (Fig.8 pan. c3).

When looking at aspect, both the automatic and human-made snow vs. ground-soil classification related to local climate. For example, they both classified more snow than grass across south slopes (between 50° and 251°), where precipitation is generally more abundant due to seasonal circulation from the Gulf of Genoa (Fig.8 pan.d1 vs. d2, see Rudari et al., 2005; Brunetti et al., 2009). On the other hand, grass classifications increased on north-facing slopes (from 250° to 351°), likely because these areas are exposed to naturally more humid conditions. Overall, the model underestimated the frequency of random-error classification along all aspects.





As for the other, less important features, they generally showed a negligible influence on the decision process. The only clear exception was relative humidity, since we found a progressive decrease of snow-classifications as relative humidity increases (Fig.8 pan. e1), coupled with an increase of grass/bare ground classification (Fig.8 pan. e2).

Finally, considering measured snow depth (by far the most important feature)), the model correctly classified all values above 400 cm as random errors, correctly matching the human classification (Fig.8 pan. h3). This is due to an instrumental limit given by the height of the sensor from the ground in this study region. Given that snow depth is the most important feature in driving the classification problem, we found a perfect match between model and human classifications (Fig.8 pan. h1 and h2).

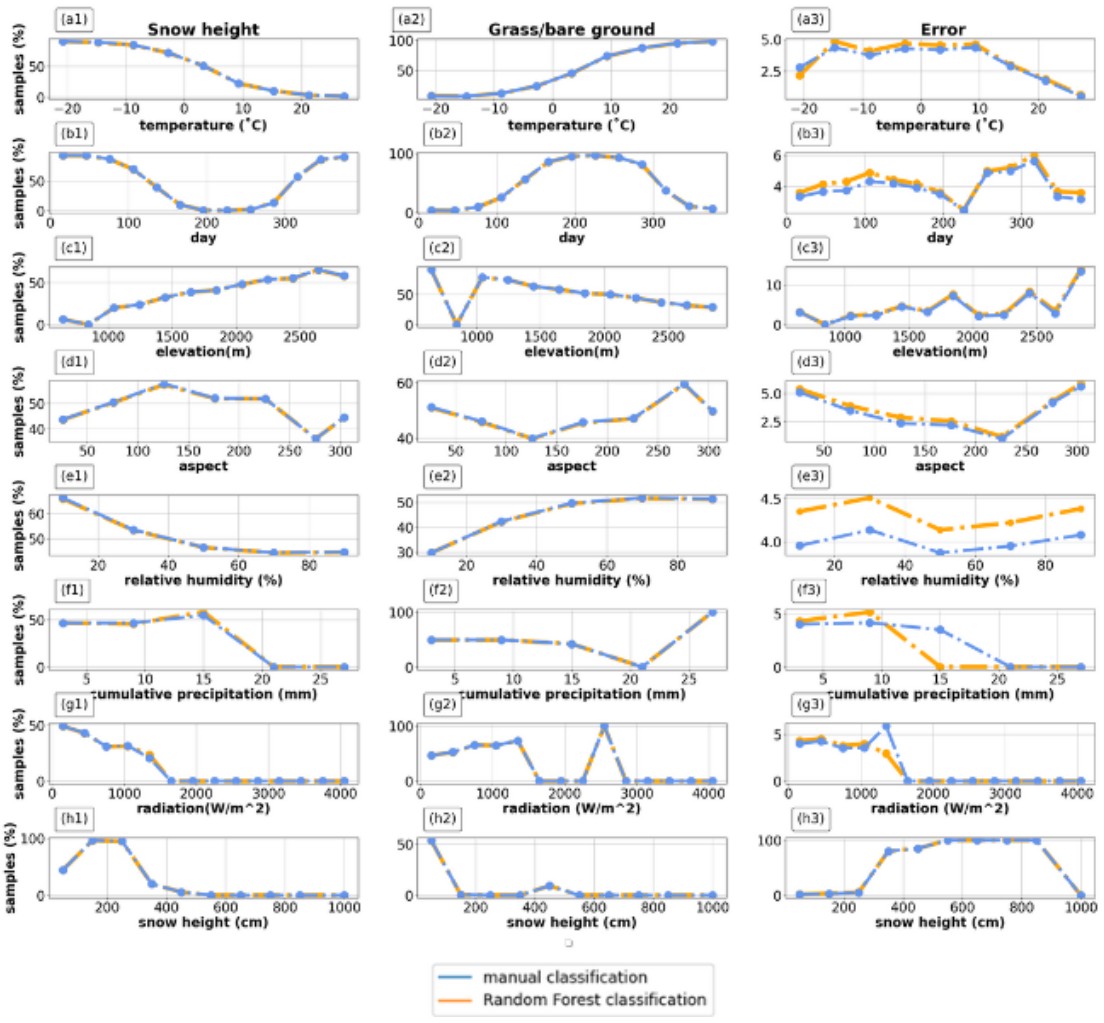

**Figure 8.** Classification results as a function of features values: left are for snow classification, center are for grass/ground classification, right are for random-error classifications. Orange is the human made classification, blue is the classification performed by the Random Forest. The x-axis reports feature values, while the y-axis reports the percentage of classification on the total. The plots refer to the test sample in Aosta valley, being it representative of the entire residual dataset. Data are normalized over the total sample size.





 ### 4.5    Validation on the rest-of-Italy sample

The application of the Random Forest on the 27 ultrasonic snow-depth sensors from the rest of Italy showed a surprising robustness in the classification of snow depth and grass/bare ground, with F1 score values between 0.93 and 0.96 across the three years. The performances of the Random Forest on the classification of both snow samples and grass/bare-ground sample proved to be comparable to the ones already noted in Aosta Valley; a severe reduction of performance was registered in the detection of random errors, for which the F1 score was below 0.05 in every year. We explain this as potentially due to the fact that we operated our own classification of this dataset, with an inevitably different subjectivity as that used by the expert forecasters in Aosta Valley; this is particularly impactful for random errors due to their smaller frequency in the sample (error sample frequency : 0.36 % in 2018, 0.92% in 2020, 0.52% in 2022).

Results of model application for two years at the exemplary station of Pratignano (Fig.10 panels a2 and b2) suggested a better performance for the model in case of higher snow depth. In other words, the model better distinguished snow from grass or bare ground when their heights were less commensurable, hence the slightly better performance in a year with higher snow depth (F1 score in 2018 : 0.95 for snow and 0.96 for grass/bare- ground; F1 score in 2022 : 0.93 for snow and 0.94 for grass/ bare-ground). This example also showed a recurring tendency to confounding snow and grass at the beginning and at the end of the season, as already noted in Aosta Valley. Considering grass classification (Fig.10 pan. a3 and b3), we also found a tendency to missclassify snow and grass during periods of intraseasonal melt. Two other examples of application of the Random Forest on exemplary sites can be found in the Appendix .We chose to show a snow-depth sensor located in the Apennines (Fig. A1) and one located in Northeast Italy (Fig. A2), to better portrait snow regimes and Random Forest performances across Italy.

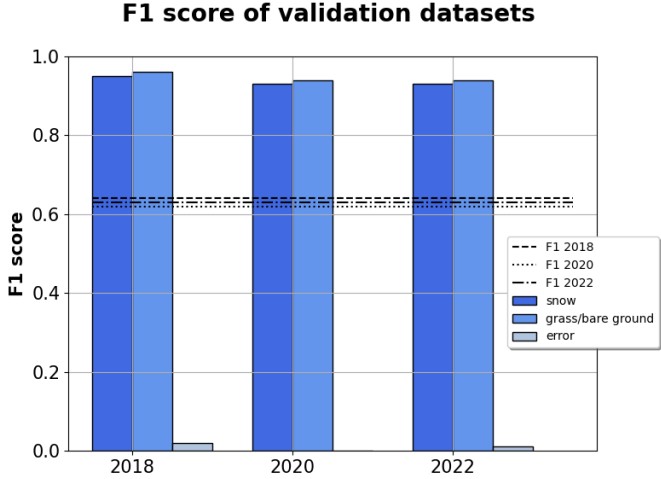

**Figure 9.** Classification performance on the 27 stations across the rest of Italy. The columns grouped along x-axis are the F1 score for snow, grass/bare ground and random-error classes, respectively, subdivided by year. The y-axis reports the dimensionless values of each scoring metrics. The straight lines are the F1 score macro averaged for each year.



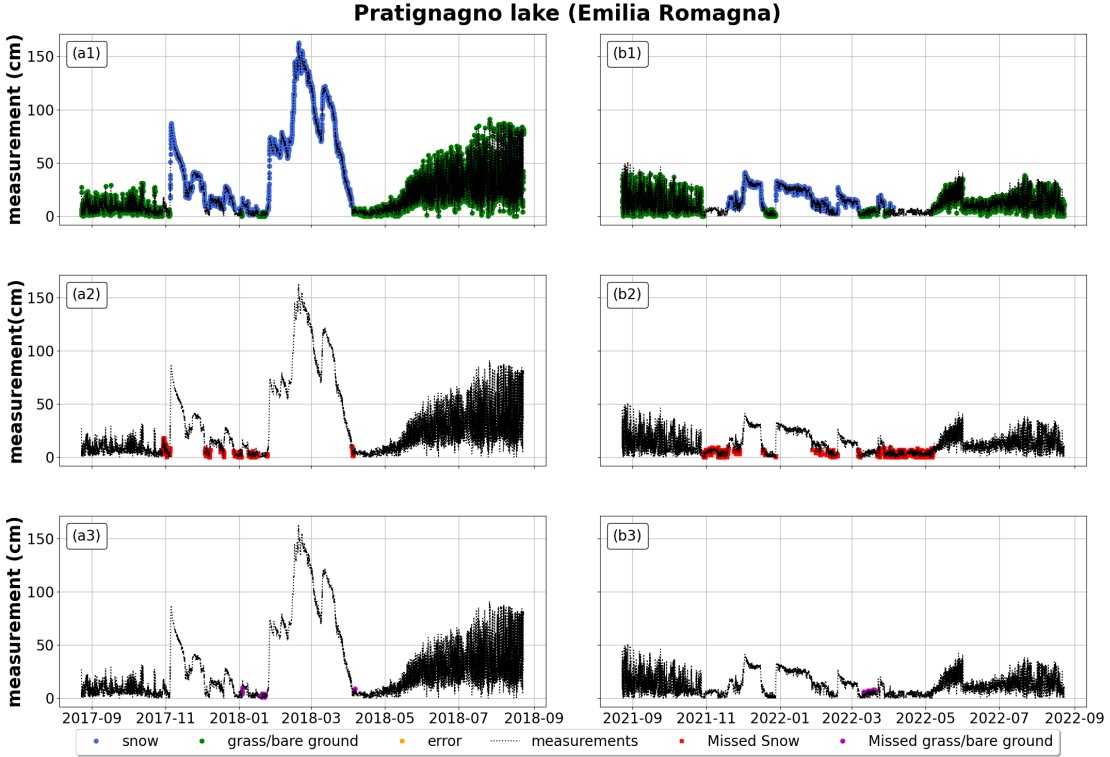

**Figure 10.** Example of application of the Random Forest on an Italian station (Lago Pratignano, Emilia Romagna). Left: October 2017 to September 2018; Right: October 2021 to September 2022. First row reports correct classification of snow, grass/bare ground, and random errors (blue for snow depth, green for grass/ground, orange for random errors); second row reports miss-classified snow depth in red; the third row reports miss-classified grass/bare ground (in purple), All plots also report measured snow depth in black (whether it represents actual snow depth, grass/ground, or random errors).

## 5 Discussion

Due to the central role that snow plays in the global water cycle (Flanner et al., 2011; Beniston et al., 2018), snow measurements

have proven to be essential in the development of trustworthy numerical-prediction models and snowpack models (Horton and Haegeli, 2022). In this framework, high-resolution measurements not only include meaningful information, for example related to snowfall intensity and amount (Lehning et al., 2002b, a) or snowmelt patterns (Malek et al., 2017; Zhang et al., 2017), but also embed a variety of noise sources that hampers their use in operations unless intensive QA/QC is performed (Avanzi et al., 2014). The overarching hypothesis of this paper was that a Random Forest classifier could replace expert manual checking and

automatically process snow-depth high-resolution measurements from ultrasonic snow-depth sensors and thus add new value to these data for hydrologic practice and research. The main findings of this paper in this regard are three.





First, the proposed Random Forest classifier was able to correctly replicate expert-made snow vs. grass/ground classifications, with F1 scores over 90% for the training/test case study of Aosta Valley. These results show that the human assessment based on expert knowledge is largely replicable (see Figure 8), at least for what concerns the classification of snow and grass/-
ground samples. While intuitively simple in nature, this differentiation is instead complex to automatize due to non-linearities across climate, snow regimes, vegetation patterns, and topography. Meanwhile, differentiating grass/ground from snow bears significant implications with regard to snow-depth assimilation in snowpack models (Bartelt and Lehning, 2002), satellite-data validation using ground-based data (Parajka and Blöschl, 2006; Da Ronco et al., 2020), and a variety of ecological analyses related to snow (Sanders-DeMott et al., 2018). In this regard, our proposed Random Forest is a pathway towards minimizing
this noise and such accelerating the use of snow depth data in science and technology by opening the way for a fast, objective, and replicable QA/QC of snow-depth data that could complement existing practice (Avanzi et al., 2014; Bavay and Egger, 2014). Regarding speed, Table 3 shows that applying our Random Forest of one season worth of data takes about 8 seconds, as opposed to an estimate of hours for visual screening based on our own experience.

| Phase | Execution time | Sample |
|---|---|---|
| Training | 00:16:29 | $1.9 \times 10^6$ |
| Testing phase | 00:02:35 | $4.8 \times 10^5$ |
| Single year validation | 0:00:08 | $2.3 \times 10^5$ |
| Visual screening | hours/days | $2.3 \times 10^5$ |

**Table 3.** Execution time.

Second, the algorithm proved to be equally robust and reliable in an independent application across the rest of Italy, at least
for what concerns the snow vs. grass/base ground classification (F1 scores above 90% for this larger domain). We explain this outcome as due to our Random Forest including all features of the Sturm and Liston (2021) snow classification, such as air temperature and precipitation, or proxies thereof (elevation for wind speed). At the same time, the vast majority of Italian sites falls between the maritime and the montane-forest snow-climatology classes, with only a small portion of tundra snow at very high, inner-Alpine elevations (Sturm and Liston, 2021). In other words, our testing sample might be quite homogeneous with
regard to snow climatology, and testing over other regions would still be helpful.

Third, we found little to none sensitivity to snow-season climatology (Figure 7), including temperature or mean snow depth. This result may point to our Random Forest being robust to different climatic regimes, including recent dry and warm snow droughts (Hatchett and McEvoy, 2018; Toreti et al., 2022; Koehler et al., 2022) and future climate change (Beniston et al., 2018). However, long-term climatic shifts will also bring about modifications to vegetation patterns (Cannone et al., 2008),
and so changes in the expected seasonality of grass vs. snow, as well as changes to the "expected" snow depth during winter (Marty et al., 2017). Both aspects will need further testing in areas with different climates.

One critical aspect of our results is the frequently reported underestimation of random errors, like spikes, particularly across the rest-of-Italy data. This may be seen as the natural consequence of our samples being inherently imbalanced towards snow or



grass/ground measurements (see Figure 3). Moreover, random errors are by definition hard to predict, with the only documented
pattern of snowflake interference within the field of view of ultrasonic snow-depth sensors (Avanzi et al., 2020b). A potential
solution in this regard is for future applications to specifically target the classification of random errors, by either including
more samples of this class or simply extending the analysis to more data. In any case, the small proportion of these errors over
the much more influential systematic issue of grass interference makes our Random Forest a promising component of future
QA/QC procedures.

**6    Conclusions**

Noise sources in high-resolution snow-depth data severely limits their automatic use in snow models, whether in assimilation
or in evaluation mode, thus affecting water management, hydrological forecasting, and emergency preparedness. In particular,
snow-depth measurements from ultrasonic sensors are prone to snow vs. grass ambiguity and random noise. Meanwhile, the in-
creasing volume of available data highlights that non-replicable, time-consuming, and error prone visual screening procedures
are increasingly less feasible. Here, we hypothesized that current practice in snow-depth data processing could be improved
by training a Random Forest to replicate expert-knowledge data processing of snow-depth data and so develop an automatic,
time-saving quality checking procedure. The algorithm used is a Random Forest classifier, a competitive and straightforward
approach compared to other machine learning algorithms. Our results show that the proposed Random Forest is reliable and
can generalize on large domains the important detection of snow vs grass/bare ground (F1 scores values above 90% even in
areas outside the original training sample). The algorithm shows little to no sensitivity to snow-season climatology, while it
is still exposed to an underestimation of rare random errors that will be subject to future studies. Our Random Forest can be
readily employed as a component of supervised or unsupervised processing procedures for snow depth.

*Data availability.* Sources of data used are reported in the paper, and include the database of the Italian Regional Administrations and
Autonomous Provinces, as accessible by CIMA Research Foundation through the Italian Civil Protection.

*Author contributions.* GB and FA designed and implemented the automatic procedure. DP, HS, and SR developed the human-made classifi-
cation procedure and provided the labelled data. All authors were involved in discussing the results, reviewing the manuscript, and drafting
the final version.

*Competing interests.* The authors declare that they have no conflict of interest.

*Acknowledgements.* We would like to thank Dr. Mirko D'Andrea for his seminal advice during the early stages of this work.





## 345 Appendix A: Random Forest Test on Italian stations

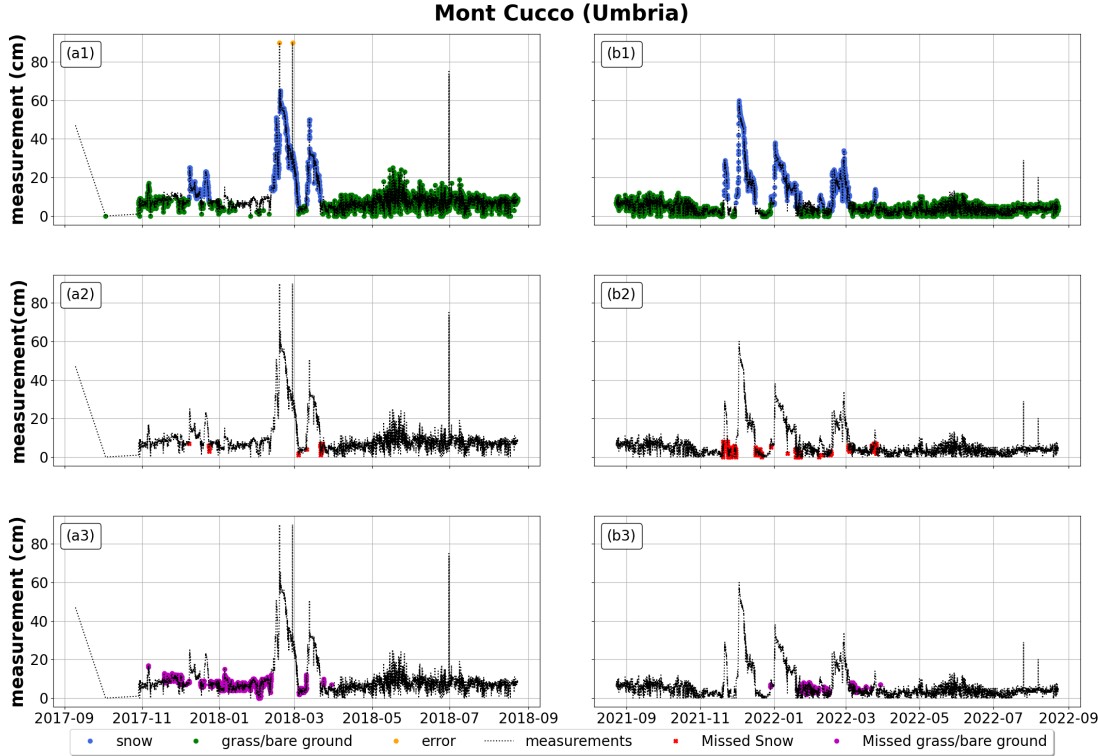

**Figure A1.** Monte Cucco ( Umbria). Application of Random Forest on an Italian station from October 2017 to September 2018 on the left, and from October 2021 to September 2022 on the right. First row reports correct classification of snow, grass/bare ground, and random errors (blue for snow depth, green for grass/ground, orange for random errors); second row reports miss-classified snow depth in red; the third row reports miss-classified grass/bare ground (in purple), All plots also report measured snow depth in black (whether it represents actual snow depth, grass/ground, or random errors)



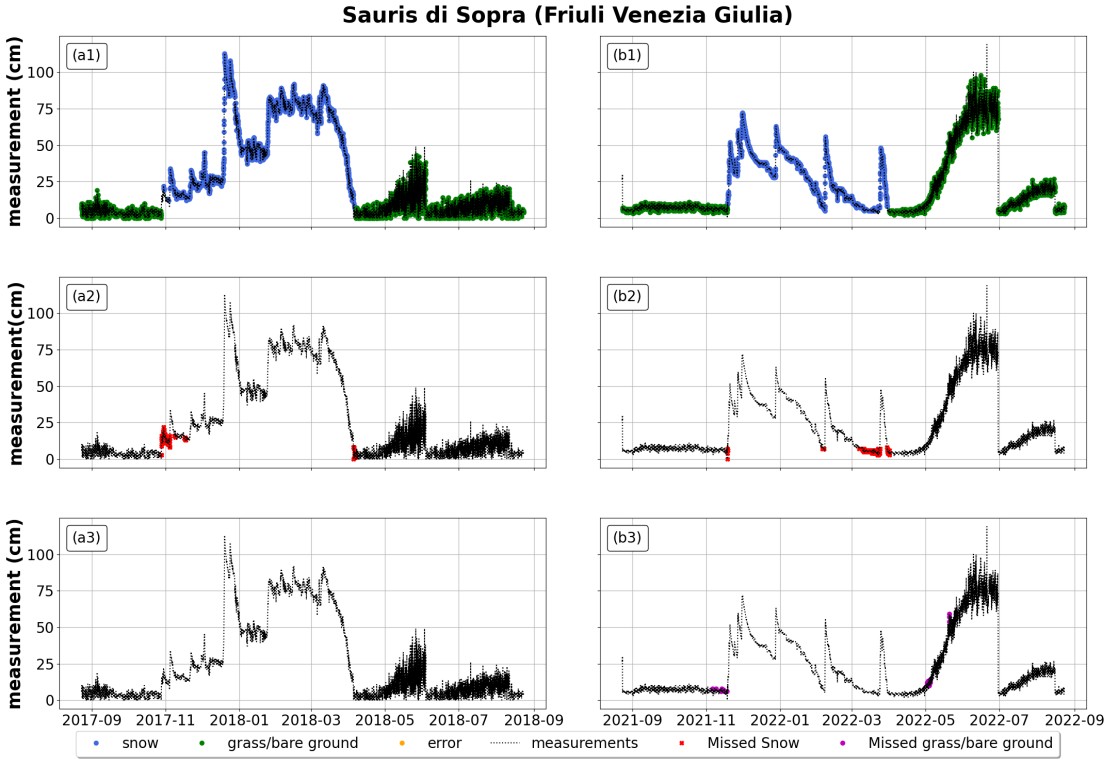

**Figure A2.** Sauris di sopra ( Friuli Venezia Giulia). Application of Random Forest on an Italian station from October 2017 to September 2018 on the left, and from October 2021 to September 2022 on the right. First row reports correct classification of snow, grass/bare ground, and random errors (blue for snow depth, green for grass/ground, orange for random errors); second row reports miss-classified snow depth in red; the third row reports miss-classified grass/bare ground (in purple), All plots also report measured snow depth in black (whether it represents actual snow depth, grass/ground, or random errors)



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
