# Peer review of "A Random Forest approach to quality-checking automatic snow-depth sensor measurements"

_EGUsphere, 2023_

## Referee Comment (RC1)

Snow depth plays a critical role in the estimation of snow water equivalent (SWE). Snow depth data from measured ultrasonic instruments are an essential part of the validation and assimilation of these models. This paper showed that a Random Forest classifier can be used to train and perform classification of snow depth from weather stations to snow/bare ground. This will help to reduce noise in SWE model coming from the misclassification of the snow depth data. In general, this a well-written paper with clear objectives, results and discussion. I recommend this paper for publication with only a few minor comments.

Specific comments:

L31-L34. This sentence is long. Consider splitting into smaller sentences.

Section 2.2. Can you give more details on how this dataset is different or not from the training set? This is not evident if you are not familiar with this region of the world.

Section 2. Do all sensors were similar ultrasonic sensors? Any effect of the different sensor types?

L116-L117. Consider adding a citation that relates snow depth to random forest. I can think of a couple.

L114. Some feature importance calculations can have a bias due to the correlation between features. This will split the importance between features (Strobl et al. 2007). Even if the feature seems uncorrelated in this case. Consider adding a sentence to acknowledge this.

Figure 4. Typo in label of graph b). "Ground" should be ground.

Reference

Strobl et al, 2007. Bias in random forest variable importance measures: Illustrations, sources and a solution. DOI : 10.1186/1471-2105-8-25

---

## Author Response (AR1)

Savona (Italy)

October 18, 2023

Dear Editor,

We would like to re-submit the manuscript A Random Forest approach to quality-checking automatic snow-depth sensor measurements to The Cryosphere.

We have revised the manuscript based on comments from the two reviewers and would like to thank you for finding the time to review our manuscript.

We confirm that all requested changes were feasible and we welcomed all of them at our best. Please find attached our point-by-point replies and the new version of our manuscript for details.

We also attached a version of the manuscript with highlighted changes in yellow. Due to technical reasons we were not able to provide a latex-diff version with tracked changes.

With our best regards,

*Giulia Blandini and coauthors*

**Reply to R1**

**Snow depth plays a critical role in the estimation of snow water equivalent (SWE). Snow depth data from measured ultrasonic instruments are an essential part of the validation and assimilation of these models. This paper showed that a Random Forest classifier can be used to train and perform classification of snow depth from weather stations to snow/bare ground. This will help to reduce noise in SWE model coming from the misclassification of the snow depth data. In general, this a well-written paper with clear objectives, results and discussion. I recommend this paper for publication with only a few minor comments.**

We thank Reviewer 1 for their constructive comments. We are happy that the Reviewer appreciated the writing style. All requested revisions are feasible and we will work in this direction as soon as the interactive discussion will be finalized.

**L31-L34. This sentence is long. Consider splitting into smaller sentences.**

Thanks for the comment. Following the suggestion, we propose this new version :

*Assessing the implication of snow-driven hydrological processes on streamflow and precipitation events helps with resource management. Indeed, a more in-depth understanding of snowmelt implications for the time and quantity of freshet supports forecasting for water management, dealing with water security and water related vulnerability. Most importantly, better understanding of snow processes enables the development of a sustainable water resource carrying capacity, which is crucial to cope with the shift in water balance caused by climate change.*

Changes: We modified **L31-L35** as mentioned above.

**Section 2.2. Can you give more details on how this dataset is different or not from the training set? This is not evident if you are not familiar with this region of the world.**

We apologize for the lack of details in describing a different dataset to an audience not familiar with the region we are working in. We added a detailed description of the geographical and meteorological differences between Aosta Valley and the rest of Italy in the 2.2 of the manuscript (see **L113-L126**). Here's the text:

*Italy ($301 \times 103\ km^2$) is a topographically and climatically complex region. Its main mountain chains, the Alps and the Apennines, are among the highest peaks in Europe. Partially snow-dominated regions like the Po river basin or the central Apennines have high socio-economical relevance (Group 2021). The Italian climate presents a considerable variability from north to south. According to the Köppen-Geiger climate classification (Beck et al. 2018), in the Alps the climate is humid and continental. Central Italy, alongside the Apennines chain, is characterized by a warm, temperate, Mediterranean climate with dry, warm summers and cool, wet winters. In Southern Italy, where the climate is still a warm temperate, Mediterranean climate, winters are mild, with higher humidity and higher temperature during*

*summer.Concerning snow-cover distribution, accumulation across the Alps is generally higher and more persistent than across the Apennines, where it is spatially more limited and more variable from one season to the others (Avanzi et al. 2023). Rivers draining from the snow-dominated Alps and a handful of basins draining from central Apennines host the vast majority of snow water resources across the Italian territory. In particular, the Alpine water basins host nearly 87% of Italian snow. The central Apennines, accumulate about 5% of the national mean winter SWE, leaving the remaining 8% – 9% scattered across the remaining basins over the territory. Intraseasonal melt, expected in a Mediterranean region, is a common feature in sites where cold-alpine and maritime snow types coexist like the Apennines (Avanzi et al. 2023).*

**Section 2.  Do all sensors were similar ultrasonic sensors?  Any effect of the different sensor types?**

To our knowledge, all snow-depth sensors operationally used in Italy are ultrasonic sensors, with an accuracy of a few centimeters (Avanzi et al. 2023). However, the implementation and everyday management of these sensors falls under the responsibility of Regional Environmental Agencies, thus only sparse information on sensor types was available for this study. Evaluating the performance of the algorithm as a function of sensor types is an interesting research question, which we will mention as a possible future development in the final section (pending availability of this information).

**L116-L117.  Consider adding a citation that relates snow depth to random forest.  I can think of a couple.**

We agree.  We will cite the work of Meloche et al. (2022), which proved the ability of a Random Forest algorithm to predict snow depth distribution from topographic parameters with a root mean square error of 8 cm (23%) in western Nunavut, Canada.

Changes: We added a citation to the work of Meloche et al. (2022) in the section 3.1 of the manuscript (see **L136-L139**).

**L114.  Some feature importance calculations can have a bias due to the correlation between features.  This will split the importance between features (Strobl et al. 2007).  Even if the feature seems uncorrelated in this case.  Consider adding a sentence to acknowledge this.  Reference : Strobl et al, 2007.  Bias in random forest variable importance measures: Illustrations, sources and a solution. DOI : 10.1186/1471-2105-8-25**

We thank the reviewer for pointing out an important aspect when dealing with Random Forest algorithms.  We acknowledge the necessity of mentioning it and we plan to add the following text to section 4.2:

*It is important to acknowledge that correlation among features and multicollinearity are problematic for feature importance and interpretation in a Random Forest. Features importance may spuriously decrease for features that are correlated with those selected as the most important (Strobl et al. 2007). On the other hand, Hastie et al. (2009) point out how that the predictive skill of the algorithm is relative robust to correlations thanks to de-correlation factors*

*involved in bootstrapping. Indeed, even low-importance features may drive the decision process of the algorithm (Avanzi et al. 2019). In our case, we chose to use all the features after testing the lack of strong correlations across features (values below -0.5 or +0.5).*

Changes: We added the above paragraph to the section 4.2 of the manuscript (see **L255-L260**

**Figure 4. Typo in label of graph b). "Ground" should be ground.**

We thank the reviewer for this remark. We will update the figure.

Changes: We corrected the figure as suggested.

**Reply to R2**

**General comment :**

The paper is well-structured, well illustrated, and easy to read. Particular care was taken to provide neat and readable figures and explanations, which I thank and congratulate the authors for. The paper's topic is both interesting and timely, in line with the increasing availability of diverse snow depth data, sometimes produced by non-professional networks or organizations, and that could serve scientific goals provided they can be qualified. The methods proposed and the analysis of the results are sound and provide a balanced evaluation of the proposed automatic quality assessment tool. I have some suggestions that I hope will help clarify some methodological points related to metrics/evaluation, and complement the perspectives. I recommend the publication of this article provided these minor suggestions are taken into account.

We appreciate the reviewer comment and acknowledge that recommended changes will improve the clarity of our work. We think all suggested modification are feasible, thus we will work towards this direction to improve our work.

**Detailed comments :**
* "what is the accuracy of a Random Forest classifier algorithm in automatically performing QA/QC of near-surface snow depth observations?" Although the choice of a RF classifiers is well justified in the paper, it seems other AI algorithm could also be used. This could be something to explore in future work. Especially, the consideration of snow height measurements as a time series and not just separate features, could possibly help identify spikes better ; this could be done through the use of AI algorithm incorporating memory features like recurrent networks or LSTM (see last point of the Detailed comments).

We agree with this suggestion and in general with the recommendation of exploring the performances of LSTM over a Random Forest. Indeed, we believe there could be possible interest in investigating the ability of a LSTM to handle time series. Although we chose a Random Forest because of its easier implementation than LSTM, we will add a reference to this in the Discussion. We will add the paragraph below:

*In recent years, Deep learning has proven successful in dealing with many complex task (Camps-Valls et al. 2021). Future research questions may investigate the ability of other algorithms in this classification problem, such as neural networks, which are able to deal with time series and incorporate memory features. One concrete example in this regard a recurrent neural networks or LSTM (Long Short Term Memory). In particular, it would be important to explore the performances of such algorithms in dealing with the recognition of the error class.*

Changes: We added the above paragraph to the section 5 of the manuscript (see **L362-L366**).

* **L149-150 "After the oversampling procedure, a sample of 1.9×10\*\*6 over-sampled measurements was used".** It is somewhat not so clear whether 1.9x10\*\*6

is the size of the total training set (including majority classes + oversampled minority class, which I believe it is) or just the oversampled minority class, which the "over-sampled measurement" in the sentence makes think. Could you be a bit more specific in the description ?

The mentioned sample size is the size of the total training set, including both the majority and the oversampled minority classes. We will modify the description to clarify.

Changes: We specified in the text that 1.9x10**6 is the size of the total training set including majority classes and oversampled minority class (see **L167-L168**).

**\* L162-165 : it should be stated that the metrics are going to be used to characterize the performance of the RF for each class separately, and then globally to characterize the multi-class performance through use of a macro-average. It may be also useful to explain the term macro-average to enhance readability.**

We thank the reviewer for their suggestion. We will clarify the meaning of macro-average and specify the use of the metrics. We will add this paragraph after L162-165:

*The metrics of precision and recall were used to characterize the performance of the Random Forest for each class separately. Then macro-averages of both measures were computed to characterize the multi-class performance. A macro average is the arithmetic mean computed giving equal weight to all classes, and is used to evaluate the overall performance of the classifier.*

Changes: We add the above paragraph in the section 3.2 of the manuscript (see **L186-L191**).

**\* L 179 : which radiation is this ? Incoming longwave, shortwave ; reflected or upcoming ones from the ground ? It should be specified because it matters for the interpretation of the importance of and relationships to this predictor. Typically, reflected shortwave radiation could be a super-help to detect snow vs grass-ground, but I assume this is not the kind of radiation that was used.**

We agree with the reviewer here, thus we will specify the type of radiation used: incoming shortwave radiation.

Changes: We specified the type of radiation (see **L198**).

**\* Both the test set and the evaluation set share years with the training set. The effect of different years on the RF performance is assessed in Fig 7 and related text, but actually, it seems to me that the temporal transferability of the algorithm (= transferability to other, completely unknown years) was not thoroughly tested within the split-sample procedure, though this is probably one key application of this algorithm. You very wisely discuss that your results "may point to our Random Forest being robust to different climatic regimes". Is there a particular reason why you did not choose an evaluation set enabling an evaluation of the RF**

**model in full spatio-temporal extrapolation mode ? Maybe related to that, how sensitive would be the performance of the RF algorithm to a moderate reduction in size of the train set, for instance to a withdrawal of the 3 complete years 2018, 2020, 2022 that would then enable an evaluation of the model in spatio-temporal extrapolation ?**

We thank the reviewer for their comments and acknowledge that more details were needed on this point.

We addressed spatial extrapolation through the validation test performed on the rest-of-Italy sample. This is because this validation test includes areas experiencing a variety of climates that are only weakly correlated with those in Aosta valley (**?**). Regarding temporal extrapolation, we preferred not to withdraw specific water years and proceed with a more standard out-of-bag validation in an effort to maximize the number of training points and climate variability in our training sample. This is particularly critical for random errors, which were the least represented class and would have been further penalized by the withdrawal of complete water years. We acknowledge that the above does not represent a full evaluation of the spatio-temporal extrapolation skills of our Random Forest and will add this consideration to our revised manuscript.Here the proposed text that will be add in the Discussion section:

*It is worth mentioning that, although the choice of validation dataset allowed for testing the spatial extrapolation abilities, a full evaluation of the spatio-temporal extrapolation skills was not achieved. The algorithm was trained on all the years available, with a standard out-of-bag validation in an effort to maximize the number of training points and climate variability in our training sample. No year was withdraw. It was aimed at reducing the impact of impoverishment of the sample on the least represented class of random errors.*

Changes: We added the above paragraph in the section 5 of the manuscript (see **L347-351**)

**Finally, the rationale behind the very short section 4.3 should be described either ahead of this section in the introduction, or within the section.**

We agree with the need for further information as suggested by the reviewer, thus We added the following paragraph in the section 3.2 of the manuscript (see **L211-215**):

*In addition to the general training strategy above, a Random Forest algorithm was also trained using the Valle D'Aosta dataset separately for each year, with 80% of the data used in training and then an out-of-bag validation with the remaining 20% of the same year data. The aim of this further test was to investigate the possible correlation between the performance of the classification by the Random Forest algorithm and annual weather characteristics. For each year, the F1 score on the test sample was analyzed against annual mean values of features used for the classification, computing correlation factors.*

**\* L 320-324 : I am not sure that the addition of more data will distinctively refine the accuracy in the "errors" class, except if you use a super huge amount of new data. I would hypothesize that other strategies may pay out with respect to this issue, and may be either explored or at least cited if you find them relevant :**

We agree with the reviewer that, despite the use of more data being likely the most straight-forward option to detect rare random errors, other options (such as other algorithms) could also be a solution.

**- maybe using other, pre-processed features could help, as for instance a Delta-HS = HS(t)-HS(t-1) with HS = Height of Snow. This could help detect unrealistic spikes or drops in snow data like the spikes remaining after RF treatment in Fig A1. This hypothesis is very basic to test.**

We believe the solution suggested by the reviewer to be a potentially effective one; moreover, the literature suggests its feasibility in operational application ( e.g. meteoIO (Bavay & Egger 2014)). Nevertheless, our aim here was to develop a fully Machine Learning procedure requiring only minimal pre-process. We would like to keep this focus for the present study. Nonetheless, we agree that coupling our Machine Learning procedure with other statistical methods could be beneficial, and we will comment on this in our Discussion section.Here a proposed paragraph:

*The use of more data is likely the most straightforward option to detect rare random errors. However, other options may prove to be effective. In light of this, the proposed algorithm may be coupled with classical QA/QC procedures imposing a-priori thresholds, like those already proposed by Bavay & Egger (2014). Such procedures could, e.g., help with the detection of spikes in data using climatological snow-depth thresholds for maximum values.*

Changes: We added the above paragraph in the section 5 of the manuscript (see **L357-361**)

**- alternatively, using AI algorithms suited to dataseries and incorporating some memory, like recurrent network or LSTM, could help if fed with snow height time-series or small extracts of them.**

We agree with the reviewer that the use of a deep learning algorithm may help in improving the performances of classification of rare random errors, requiring however further study. We will mention this as a future opportunity in the Discussion as explained in our answer to the first comment above.

**- finally, have you considered the use of webcam images from nearby the sta-tions within the same elevation/aspect, that could provide a simple, maybe not completely reliable snow-nosnow information, but with errors maybe not com-pletely correlated with the RF errors ?**

We believe the suggestion made by the reviewer could be interesting in specific research settings, especially because coupling different data types and data sources may enrich the algo-rithm performances (Karpatne et al. 2018). At the same time, webcams are not systematically installed at operational, automatic weather-snow stations in Italy and elsewhere, which signif-icantly limits the applicability of this approach in the real world.

**Edits :**
**L5-6 : "with particular reference to differentiate snow cover from grass or bare ground data and to detecting random errors (e.g., spikes)" -¿ to detect ?**

Changes: We correct *to detecting* into *to detect* (see **L5-L6**)

**L54 : "It is clear then the necessity for a quality checking procedure, that ought to... " it seems there is a syntax issue**

Changes: We modified the sentence sintax as follow :  *It is clear then the necessity for a quality checking procedure that ought to be defined ...* (see **L55-L56**).

**Fig 2 : adding the contours of Italy would be nice**

Changes: Done.

**L143 : end,askowleding**

Changes: We correct the grammar error (see **L160**)

**L143 : " the work of (Ponziani et al., 2023) in which no clear evidence of out-performance of any strategy, " It seems some words are missing**

Changes: We completed the sentence as follows: *To this end, aknowledging the work of (Ponziani et al., 2023) in which no clear evidence of out-performance of any strategy is shown, ....* (see **L160-L161**)

**L 162 : "precision(measure of"**

Changes: "precision (measure of" (see **L179**).

**L 208 : I guess a "." is missing before "Fig 5".**

Changes: We Corrected it.

**Caption of Fig 5 : "model.In".**

Changes: We Corrected it.

**Fig 8 : maybe use the same vertical scale across rows, as the amplitudes are otherwise quite hard to compare esp. in the 3rd column.**

Changes: We used the same vertical scale for the first two column. We choose to use a different scale along the third column to handle the imbalance of the dataset.

**References : there is an issue with the Avanzi et al 2020, 2021 and 2022 references that are always stated twice.**

We thank the reviewer for their comments. We will modify the text accordingly.

Changes: We solved the issue.

**References**

Avanzi, F., Gabellani, S., Delogu, F., Silvestro, F., Pignone, F., Bruno, G., Pulvirenti, L., Squicciarino, G., Fiori, E., Rossi, L. et al. (2023), 'It-snow: a snow reanalysis for italy blending modeling, in situ data, and satellite observations (2010–2021)', *Earth System Science Data* **15**(2), 639–660.

Avanzi, F., Johnson, R. C., Oroza, C. A., Hirashima, H., Maurer, T. & Yamaguchi, S. (2019), 'Insights into preferential flow snowpack runoff using random forest', *Water Resources Research* **55**(12), 10727–10746.

Bavay, M. & Egger, T. (2014), 'Meteoio 2.4.2: a preprocessing library for meteorological data', *Geoscientific Model Development* **7**(6), 3135–3151.
**URL:** *https://gmd.copernicus.org/articles/7/3135/2014/*

Beck, H. E., Zimmermann, N. E., McVicar, T. R., Vergopolan, N., Berg, A. & Wood, E. F. (2018), 'Present and future köppen-geiger climate classification maps at 1-km resolution', *Scientific data* **5**(1), 1–12.

Camps-Valls, G., Tuia, D., Zhu, X. X. & Reichstein, M. (2021), *Deep learning for the Earth Sciences: A comprehensive approach to remote sensing, climate science and geosciences*, John Wiley & Sons.

Group, T. W. B. (2021), 'Italy-Climatology', `https://climateknowledgeportal.worldbank.org/country/italy/climate-data-historical/`. Accessed: 2023-09-15.

Hastie, T., Tibshirani, R., Friedman, J. H. & Friedman, J. H. (2009), *The elements of statistical learning: data mining, inference, and prediction*, Vol. 2, Springer.

Karpatne, A., Ebert-Uphoff, I., Ravela, S., Babaie, H. A. & Kumar, V. (2018), 'Machine learning for the geosciences: Challenges and opportunities', *IEEE Transactions on Knowledge and Data Engineering* **31**(8), 1544–1554.

Meloche, J., Langlois, A., Rutter, N., McLennan, D., Royer, A., Billecocq, P. & Ponomarenko, S. (2022), 'High-resolution snow depth prediction using random forest algorithm with topographic parameters: A case study in the greiner watershed, nunavut', *Hydrological Processes* **36**(3), e14546.
**URL:** *https://onlinelibrary.wiley.com/doi/abs/10.1002/hyp.14546*

Strobl, C., Boulesteix, A.-L., Zeileis, A. & Hothorn, T. (2007), 'Bias in random forest variable importance measures: Illustrations, sources and a solution', *BMC bioinformatics* **8**(1), 1–21.